# High-Tech Classroom Management: Effects of the Use of an App on Disruptive and On-Task Classroom Behaviours for Students with Emotional and Behavioural Disorder

**DOI:** 10.3390/bs13010023

**Published:** 2022-12-26

**Authors:** Gabriel Cohen, Neil Martin

**Affiliations:** 1Department of Special Education, Oranim Academic College of Education, Kiryat Tiv’on 3600600, Israel; 2Behavior Analyst Certification Board, Littleton, CO 80127, USA

**Keywords:** high-tech classroom management

## Abstract

Students with emotional behavioural disorders may exhibit extremely challenging behaviour that interferes with their academic achievement and social relationships. Failure at school frequently leads to a succession of poor life outcomes including increased rates of unemployment or underemployment. Increasing on-task behaviours and decreasing disruptive classroom behaviours is of crucial importance. If successful, this may promote positive experiences and outcomes in terms of effective learning, and, ultimately, greater opportunities in life. This study evaluated a high-tech approach to classroom management using an App* that offers elements of choice and predictability to students. Teachers were provided with two hours of training on how to upload lesson plans to their smartphone and how to broadcast onto screens in classrooms. A multiple-baseline design across four participants was used and the data suggested that the use of the App resulted in both increases in on-task behaviour and a reduction in disruptive classroom behaviour for all participants.

## 1. Introduction

Emotional and Behavioural Disorder (EBD) refers to emotional, behavioural or psychiatric disorders, including attention deficit hyperactivity disorder, depression, and mood or anxiety disorders [1,2]. Typically, individuals with EBD are characterised by both internalising behaviours, such as fear and anxiety, and externalising behaviours, such as aggression and vandalism. The former includes symptoms of depression, anxiety, social withdrawal, unhappiness, fear, isolation, phobias and low self-esteem, whereas the latter includes conflicts with others, delinquency, hyperactivity, disruptive behaviour and aggression [3,4]. In educational environments, students with EBD may exhibit extremely challenging behaviour that interferes with academic achievement and social relationships [5]. They present with a host of poor academic, social, emotional and post-school outcomes, and higher rates of disruptive and off-task classroom behaviours than typically-developing individuals [6]. Researchers have also found a tendency toward course failure, lower average grades and absenteeism; the latter was found to be strongly linked to poor academic outcomes, disruptive behaviours and aggression [7,8]. In turn, disruptive behaviours have a negative impact on teachers’ instructional time and impede classroom learning, making it less likely that students will succeed academically [9]. Students with EBD are also more likely to experience disciplinary exclusion [5,10,11], and are at greater risk both of being placed in more restrictive settings than students with other disabilities [10] and becoming involved in the criminal justice system [12]. Failure at school frequently leads to a succession of poor life outcomes, including increased rates of unemployment or underemployment [13,14]. Longitudinal research reports that more than 50% of students with EBD dropped out of school, and fewer than 50% of those who remained at school graduated with a diploma. Moreover, 20% were arrested at least once before they left school, and more than 50% were arrested within a few years of leaving school. Among those who dropped out, 70% were arrested [15]. In this context, increasing on-task behaviours and decreasing disruptive classroom behaviours for students with EBD is of crucial importance as a proactive strategy. If successful, this may promote positive experiences and outcomes through effective learning [16] and additionally decrease teacher attrition and high turn-over, ultimately enhancing the quality of learning and the quality of life of students, their families and educational personnel. 

School-wide Positive Behavioural Support (SWPBS) [17] is implemented by more than 20,000 schools in the United States and internationally [18,19]. Its primary aim is to address social and behavioural concerns in schools. A substantial body of literature demonstrates the impact of SWPBS on reducing disruptive behaviour and improving appropriate on-task classroom behaviours [20,21]. SWPBS is also seen as promoting academic achievements and effective classroom management [22,23]. By definition, effective classroom management is the process of organising and conducting a classroom in order to enhance learning, on-task behaviours and life quality [24,25] identified effective classroom management as an essential teaching skill and suggests that effective teachers minimise disruptive behaviour and enhance on-task classroom behaviours and learning environments that allow for students’ intellectual and emotional growth. [26] believed that classroom management encompasses all that a teacher does to organise students, space, time and materials so that student learning can take place. [27] suggested that classroom management involves teacher actions and instructional techniques to create a learning environment which facilitates and supports active engagement in academic, social and emotional learning. In addition, several empirical studies have documented the effect of SWPBS on students with, or at risk of, EBD [28]. These studies were conducted in a variety of settings, including elementary schools [8,29,30], secondary schools [31], public schools [32,33], and alternative education settings [34]. To date, such research has used single-case designs [35,36]. Generally, many of the studies conducted – most of which used proactive, effective, multi-component interventions – have provided empirical evidence that SWPBS has a positive impact on social and academic behaviours for students with or at risk of EBD [32,33].

Despite the evidence for the use of common multi-component classroom interventions, their effectiveness may be limited due to two primary considerations: the high costs involved, and the number of skills that teachers need to master for this endeavour within a short period of time relative to years of acquiring and practising other teaching skills [37]. In a recent study, Hickey and associates [38] investigated the efficacy of a universal multi-component classroom management training intervention. Amongst other elements, the intervention included the establishment of rapport, providing clear rules and instructions, giving attention to appropriate and adaptive behaviour, and ignoring disruptive classroom behaviours. In addition to examining the impact of the multi-component classroom intervention on teachers’ and pupils’ behaviour, [38] also assessed its affordability and feasibility. They found that, despite the knowledge gained within the field, teachers frequently reported difficulties in coping with disruptive behaviours that arose in the classroom. Furthermore, many lacked knowledge of research-based strategies and practices which can promote positive learning environments [39,40,41]. Moreover, the researchers found that this type of intervention was not as effective as they expected for either pupils or teachers. In addition, the intervention proved to be very costly (involving training time, travel and accommodation costs), making it unaffordable and not feasible to implement more widely.

Brady, Padden and McGill [42] explain that multi-component classroom management programmes may only be effective when they are also affordable and easy to use. Many schools may not be willing or financially capable of implementing lengthy multi-component programmes, particularly with the staggered introduction inherent in some research designs, which requires additional resources, and delays on-site training and teaching [43]. High-tech classroom management aims to encompass the accessibility, affordability and feasibility of effective classroom management intervention programmes through the utilisation of technology. In recent years, we have witnessed the development and improvements in information technology, the integration of communication networks and the growing use of mobile digital media (consisting mainly of smartphones and tablets) which have permeated various aspects of learning in an affordable manner [44]. Due to the advantages of portability, the ability to access information at any time and inherent affordability, mobile learning tools are being used increasingly for high-school teaching [45]. 

Since 2007, mobile phones using Apple’s iOS or Google’s Android systems have taken over the mobile market. The latest figures show an increasing number of smartphone users year after year. In 2022, the number of global smartphone users is estimated at 6.6 billion marking a 4.9 percent annual increase [46]. Such accessible and commonly used devices are attracting software developers to design third-party mobile applications. Third-party mobile applications (Apps) are software programs that expand the utility of smartphones [47]. In May 2013, Apple celebrated its 50 billionth App download, with Google trailing only slightly behind with 48 billion [48]. This new App market has resulted in over $9bn being paid to developers for Apple Apps alone. In recent years, increasing numbers of teachers appear to be using high-tech classroom management Apps, in order to make interventions not only effective but–more importantly–affordable and feasible [49,50,51].

The research literature suggests that the elements of choice and predictability within the classroom promotes attentiveness and learning [52,53,54,55]. Choice Theory is posited as beneficial to solving classroom management problems, based on the notion that teachers cannot control the behaviour of their pupils simply by telling them what to do but can play a critical role in helping students to make choices, resulting in positive behavioural changes [56]. Glasser’s Choice Theory [57] has influenced teachers’ classroom management, creating environments and curricula that cultivate appropriate, on-task behaviour through meeting student needs for a sense of belonging and empowerment [58]. Glasser conducted manipulations of choice in educational environments and found that choice strengthened the perceived sense of control and helped to reduce anxiety and fear, which are known to diminish attentiveness and learning [59]. Evidence from neuroscience suggests that, when we feel threatened, the prefrontal cortex, the part of the brain linked with learning, shuts down [59]. Individuals who are exposed to uncertainty (the opposite of predictability) are likely to feel threatened, which increases negative emotional reactions such as fear and anxiety [59]. These are the two primary characteristics of students with EBD [1,4], who tend to present with higher rates of disruptive and off-task classroom behaviours than typically-developing individuals [6]. Such students may experience anxiety during lessons in which they cannot clearly predict the lesson structure or the amount of time allocated to each task, or do not understand the topic aims and objectives, and this, in turn, contributes to reduced attentiveness [59].

The general concept for the development of an App for classroom management was to embed elements of choice at various points during a lesson, and to provide a visual representation of the progress of the lesson to make the lesson predictable, all of which was visible to students on a large flat screen within the classroom. The App was designed in such a way that teachers needed to embed two choice points during each lesson. Both the choices and the point at which they became available during the lesson were chosen by the teachers. For example, the second segment of a lesson might have been ‘focused work’, and the teachers would offer the students the choice between individual or group work. During the course of the lesson, students would see an image following a path (e.g., a dotted line) to indicate the point reached (and elapsed time) and remaining course of the lesson (and remaining time). The image continued to move throughout and in parallel with the course of the lesson with specific key markers reached along the way. Additionally, appropriate on-task behaviours, for example, raising a hand for help or engaging in quiet discussion were represented by appropriate icons which also appeared on the screen at the relevant point. Teachers prepared their lesson as they would normally and then used the App to upload the key elements of the lesson to their smartphone, a process that took approximately 90–120 s. In the classroom the teachers would then broadcast the visual representation of the lesson to the classroom’s flat screen. In the event of unforeseen circumstances or time-keeping issues, the teacher could quickly make changes via the App and update the screen. For example, they could shorten, or adjust, remaining sections to ensure that the lesson ended precisely on time and that the students were also aware of the new expectations. 

This study aimed to evaluate the use of the App in terms of both increasing on-task behaviour and reducing disruptive classroom behaviours in students with EBD using a multiple baseline across subjects design [60]. The justification for the designing of the App is two-folds; to enhance students active participations during lessons and reduce related anxiety and stress. Further description and characteristics of the App appear in Section 2 below.

## 2. Method

### 2.1. Participants

This research consists of a specific case study of four students. Participants were selected based on teachers’ referral of students with high levels of disruptive behaviours and low levels of on-task classroom behaviours during regular classroom instruction. Four teachers in a small, public Special Education school for students with EBD were each asked to select one of their pupils for inclusion in the study, resulting in the selection of four male participants (Oscar, Simon, Mark and Richard), aged between 11 and 12 years, one from each of four classes of ten students. All four participants were reported to be in the low-average range of intelligence. The four teachers who participated in this study had the minimum of a first degree, a teaching qualification, and were employed by the Ministry of Education in Israel and they had been teaching in the school for a minimum of six years. Teaching assistants who were blind to the purpose of the study collected data on participants’ on-task and disruptive behaviours and teachers’ social contact with the participants during the different phases of this study (one assistant for each participating student). All assistants had at least two years’ experience in data collection and the implementation of behavioural interventions. Information about the purpose of the study was provided to the participants, their families, their teachers and the Director of Special Education and consent was obtained from all in writing. In addition, the Director of Special Education found no conflict of inteterst, and all ethical issues were resolved. 

### 2.2. Setting

The study was conducted in a small regional school for Special Education located in the north of Israel. All sessions were conducted during the first lesson of the day (0800-0840) and within the participants’ main standard-sized classrooms (measuring approximately 17–25 m^2^) Class schedules and lesson content, as planned by the teachers, remained constant across all phases of the study.

### 2.3. Materials

An App was specifically designed and developed for the study. The App was used both for teachers to upload lessons in segments to their smartphones and to broadcast the lesson to a flat screen in the classroom, so two separate interfaces were designed, one for the teachers and one for the participants. The dual-interface App was developed on the Swift platform using the iOS SDK framework, X-Code, and Cocoa Controls programmers’ application tools. For the teachers’ interface, it was critical for the App to be genuinely user-friendly, requiring only a few clicks to upload entire lessons, and a single click to broadcast a lesson at any location within the school, and all in less than two minutes. It was also important to design the App in such a way as to ensure that (a) it was energy efficient and did not drain the mobile phone battery, (b) it had error- and bug-detection capabilities, so that potential viruses would not slow or freeze the mobile phones, and (c) allowed only the content from the App to be broadcast on the screen. The screen also changed at regular intervals, showing different screens at different times. For the first five minutes of each lesson, the screen showed the entire layout of the lesson with its segments and alternative choices, their timing and expected on-task behaviours. Then the screen would zoom in to focus on the current segment or alternative, with its timing and targets. After two minutes, it would zoom out, to illustrate the entire layout of the lesson for 30 s. This cycle continued until five minutes before the end of the lesson. Then the screen showed the current and follow-up segments with all their components.

### 2.4. Teacher Training

Teacher training was conducted for precisely 120 min for each of the four teachers in seven steps. For the first step (approximately five minutes), the teachers were introduced to the App and its components on the researcher’s smartphone and the researcher explained each component. Step two (also approximately five minutes) involved the researcher demonstrating how to upload a lesson using the App, and each of the teachers practised uploading a lesson using the researcher’s smartphone while the others watched. In step three the researcher provided positive or corrective feedback where necessary and answered any questions for approximately another ten minutes. In step four, the researcher provided the teachers a username and password, enabling them to download the App onto their own smartphone and practise in their own time. Step five occurred on the following morning before lessons began. The researcher provided each teacher ten minutes introduction explaining the procedures. The researcher monitored the teachers as they uploaded their lessons on to their respective smartphones, and broadcast to the screen in their respective classrooms. In step six, a few minutes before a lesson began, the participants were shown by the researcher the overall lesson plan on the screen, including the segments and choices, the time for each segment and the expected on-task behaviours. Participants were also shown by the researcher the progression of the lesson as it appeared on the screen and questions were answered. This step lasted approximately ten minutes per classroom. In the final step, the teachers were instructed and explained by the researcher to begin teaching the lesson they had planned and uploaded onto the App and the lesson was monitored to ensure synchronisation of the lesson with the screen content, with adjustments made to the content via the App accordingly. The lesson commenced as soon as the teacher clicked ‘Run’ to begin the clock progression on the screen.

### 2.5. Experimental Design and Dependent Variables

A multiple baseline experimental design across participants was used for this study. The target behaviours were various on-task and disruptive classroom behaviours. These were selected because they were those typically identified as targets for change with students with EBD [61] and they have been consistently used in the literature as behavioural indicators that are highly relevant in the educational environment [62,63,64,65]. Specifically, on-task behaviours were based on previous research conducted by Moore et al. [66,67]. Disruptive behaviours were identified from previous work by [68] and [69]. In order to promote the cooperation of all those involved, and for increased social validity, on-task and disruptive classroom behaviours were operationally defined jointly by the researcher, participants and teachers. The on-task and disruptive behaviours are listed in Table 1. These were defined during a meeting with participants and teachers ensuring the voice of the four participants is heard and highlighted. 

### 2.6. Data Recording System

20-min observations of 45-min lessons were conducted in the participants’ classrooms five times each week commencing at a quasi-randomised time between 0805 and 0825. Although the lesson began at 0800, the earliest data collection could commence was 0805 to allow for punctuality and time-keeping issues. Data collection was restricted to 20 min to acknowledge that the assistants had other roles and responsiblities to perform and so needed to be attainable and able to be sustained for several weeks. To ensure that data collection period provided a cross-sectional representation of the entire lesson, the data collection start times were quasi-randomly determined by asking the assistants to select one of five slips of paper (labelled ‘0805’, ‘0810’, ‘0815’, ‘0820’, ‘0825’) from a box before the class commenced.

Occurrences of on-task and disruptive classroom behavior were recorded using partial interval recording with 10-s intervals. If the participant exhibited any of the on-task or disruptive behaviours at any time within a ten-second interval, the interval was scored as an occurrence. The percentage of intervals with on-task or disruptive behaviours was calculated by adding up the scored intervals for each, dividing by the total number of possible intervals minus any missed observations, and multiplying by 100. 

In addition, teachers’ social contact with the participants was also recorded. Differential social contact during the various phases of the study could likely influence the occurrence of the dependent variables and so was required to further enable the assessment of behaviour change in participants and the potential of a transactional relationship caused by changes in social contact. Social contact was broadly defined as any contact initiated by a teacher towards a participant, whether physical, for example, a stroke or a pat on their shoulder, or verbal, for example, ‘well done’, ‘good work’ or a reprimand. Social contact was recorded as either social contact that appeared to have have the function of providing positive reinforcement to increase behaviour (*social^+^*) or social contact that appeared to be functionally punitive in nature to decrease behaviour (*social^−^*).

The assistants received specific training in partial-interval data recording using simulations and mock observations. The data collectors completed their training once 90% inter-observer agreement (IOA) had been reached with the researcher. To protect against observer drift, the researcher reviewed the operational definitions of the target behaviours and social contact with the assistants on a weekly basis. 

### 2.7. Inter-Observer Agreement

The researcher or another data collector independently observed 33–39% of all observation periods. Agreement was recorded when both observers recorded an occurrence or non-occurrence within a given interval. IOA scores for on-task, disruptive, and teacher social contact behaviours were then calculated on an interval-by-interval basis by dividing agreements by the total number of agreements plus disagreements, multiplied by 100 [60]. All IOA scores for all particpants across all recorded behaviours were acceptable with a range of 82.5% to 95.0%.

### 2.8. Social Validity

For the purpose of this study, the researcher modified a feedback questionnaire previously developed by Sasso, Reimers, Cooper, Wacker, Berg, Steege, Kelly and Allaire [70]. Modifying pre-used social validity questionnaire is essential because studies use various interventions and impact differently on participants and the behaviour change they experience. Essentially, social validity refers to the extent in which the behaviour change efforts impact favorably upon participants. Researchers utilize variety of social validity methods in their studies and identify three primary focuses of social validity assessment: the social importance of the intervention goals; the acceptability of intervention to relevant participants; and the social importance of treatment effects [71]. The ten-item questionnaire is illustrated in Table 2. It was used to ask participants, teachers, teaching assistants and parents to rate their experience and level of satisfaction with the use of App on a 1–5 rating scale, with higher values indicating strong agreement. The questionnaires were completed anonymously, with a return rate of 100%. Modifications to the original social validity questionnaire focused on asking questions specific to the intervention used in this study.

### 2.9. Intervention Phases

*Baseline*: Participants and teachers were observed and data recorded (as described above) during a typical first lesson at the school. The stability of the baseline data were used to determine the first participant to start the intervention.

*Intervention*: The intervention was the introduction of the use of the App (applied first to Oscar as his baseline data appeared the most steady and stable). All students were instructed to attend as usual and to look at the screen periodically to be reminded of the lesson progress, segments, choices and appropriate behaviours. This was replicated for each participant based on the protocols of using a multiple baseline across subjects design.

*Follow-Up*: The follow-up phase was identical to the intervention phase except that data were only collected very intermittently to see if the intervention was still having an effect with minimal input from the researcher.

### 2.10. Treatment Fidelity

To ensure the integrity of the implementation of the intervention, each teaching assistant also monitored 25% of the intervention lessons using a simple check-list. Each lesson had to be a maximum of 45 min long, two segments had to include two alternatives, each segment of the lesson had to specify one or two on-task behaviours, the display had to show an image moving along a dotted line (to simulate the progression of the lesson), the teacher had to be synchronised with the content displayed on the screen. Treatment fidelity was 100%.

## 3. Results

Figure 1 shows the on-task and disruptive classroom behaviours of the four participants during baseline, intervention and follow-up phases. All four participants made gains in on-task behaviours and had some reductions in disruptive behaviours following the introduction of the App. The most striking effects in terms of both increases to on-task behaviour and reductions in disruptive behaviour were seen for Richard, with gains/losses occurring at a more modest but steady pace for the other participants. 

Comparison of the levels during baseline and intervention observations for the participants shows a mean increase of 30% for on-task behaviour (baseline mean = 16%, range = 0–27%; intervention mean = 46%, range = 19–86%) and a decrease of 33% for disruptive classroom behaviours (baseline mean = 85%, range = 71–100%; intervention mean = 52%, range = 11–83%).

The data for Oscar suggest a steady, stable and modest intervention effect for on-task and disruptive classroom behaviours. The difference between the mean levels during baseline and intervention was +19% for on-task behaviour (baseline level = 15%, intervention level = 34%) and −16% for disruptive classroom behaviours (baseline level = 87%, intervention level = 71%). The steadily increasing trend in on-task behaviours seems to correspond with the declining trend in disruptive classroom behaviours. Oscar continued to increase his on-task behaviours and show decreases in his disruptive classroom behaviours during the final follow-up phase. Similarly, Simon’s data also suggest a moderate yet consistent effect of the intervention on both on-task and disruptive classroom behaviours. The difference between the mean levels during baseline and intervention was +15% for on-task behaviour (baseline level = 24%, intervention level = 39%) and −21% for disruptive classroom behaviours (baseline level = 83%, intervention level = 62%). The follow-up phase for both dependent variables suggested that Simon continued to increase his on-task behaviours consistently (*M* = 74% during follow-up) and decrease his disruptive classroom behaviours (*M* = 30% during follow-up).

Mark’s data depict a more variable effect for both on-task and disruptive behaviours. The difference between the mean levels during the baseline and intervention was +36% for on-task behaviours (baseline level = 17%, intervention level = 53%) and for disruptive classroom behaviours, the difference was −38% (baseline level = 77%, intervention level = 39%). The clear and sharp upward trend in on-task behaviours corresponds to a downward trend in disruptive classroom behaviours. The follow-up data suggest that Mark continued to increase his on-task behaviours and decrease his disruptive classroom behaviours during follow-up.

The on-task and disruptive classroom behaviour results for Richard show relatively immediate and consistent intervention effects for both behaviours. The difference between the mean levels of on-task behaviours during baseline and during intervention was +51% (baseline level = 5.8%) and for disruptive classroom behaviours −60% (baseline level = 94%) with no overlap for either dependent variables. The sharp increase in on-task behaviours appears to correspond to an equally sharp decrease in disruptive classroom behaviours. This effect was maintained during follow-up. 

Teachers’ social contact, per session, is shown in Figure 2. The mean level of *social^+^* contact for all four teachers across all phases was low but highly stable. *Social^−^* contact was 71% at baseline and 40% during the intervention, a mean decrease of 31%, with a further decrease to 20.3% during follow-up.

### Social Validity 

The results from the feedback questionnaire from each of the participants, teachers, teacher assistants and parents (n = 16) are shown in Table 3. The responses provide an encouraging evaluation with combined responses indicating that the use of the App was acceptable and appealing with minimal risks involved in terms of using it. The overall satisfaction score was 95.25%.

## 4. Discussion

A major training element of the study involved technology: making use of smartphones, screens and an App. The use of technology is contemporaneous with the current era, which has changed dramatically and quickly over recent years and for which, as a consequence, there is a paucity of research. However, some recent research has suggested that high-tech methods of teaching new skills are effective, economical and simple to apply and often result in positive outcomes [44,47,49]. The current study made extensive use of App development technology to make learning, as well as teaching, effective, accessibly, easy and, ideally, motivating and fun. The App appeared to be popular and willingly accepted by participants, teachers, teacher assistants and parents, and anecdotally by other school personnel, enhancing its social validity. Two parents requested to use the App at home for their child’s evening routines (e.g., a homework session from 1800–1830 with a choice of where to do this; meal time from 1830–1900 with a choice of dessert, etc.). Some other school personnel requested use of the App as they were delighted with the positive responses of students.

The data collected at baseline for each participant’s on-task and disruptive behaviours were used to set specific outcome criteria for posthoc analysis of success [72,73]. For Oscar, Mark, and Richard, their on-task behaviour outcome criterion was set at 65%. For Simon, his on-task learning behaviour outcome criteria was 75%. Criterion for the outcome for disruptive classroom behaviours was set at 30% for all four participants. For on-task behaviours, Mark achieved the target criterion in the last five of 14 intervention sessions (67%, 79%, 83%, 85% and 84% respectively). Richard achieved the target criterion in five of his 13 intervention sessions including the last four (65%, 79%, 82% and 86%). Whilst not reaching their set targets, the remaining two participants also achieved relatively high on-task scores with ascending trends. For disruptive classroom behaviours, both Mark and Richard achieved the target in six of their intervention sessions. Participants Oscar and Simon, although not reaching their targets, also showed descending trends. 

Visual analysis of the data recorded for three of the four participants reveals what appears to be only a gradual behaviour change from baseline to intervention phases for both on-task and disruptive classroom behaviours (see Figure 1) and may perhaps hint at weak experimental effects. Engel and Schutt [74] identified five limitations of using visual interpretations for single-subject design findings, the second of which relates to the immediacy of the behaviour change–when the change is not evident immediately after the intervention is presented, this may negatively influence the experimental effects. The multiple baseline design across four individuals in four separate classes with four different teachers, classes and lessons, adds credibility to the probability that the results, albeit showing only gradual change from baseline to intervention in three cases, are related to the application of the intervention. Additionally, the teacher-participant social contact data (Figure 2) shows no change to *social^+^* contacts which could have been an alternative explanation for the increase in on-task behaviour. Indeed, the clear decreasing trend for *social^−^* contacts suggest that the use of such contact was also reduced as a consequence of the intervention which led directly to decreased disruptive behaviour and so reduced the necessity for such contact.

The research literature is replete with suggestions on how to enhance quality of life for students at all ages, including through empowerment with choice and a degree of control [24,58,75]. Many related investigations have evaluated multi-element behavioural training programmes, some of which were highly effective and resulted in notable behavioural changes, yet which often required excessive resources and many training hours for multiple highly skilled personnel [73,76]. Such interventions may be beyond the reach of educational providers with limited funds. The results from this study contribute to this body of research anad offer a simple and cost-effective alternative to classroom management. With at, it must be stated that the intervention was applied first thing in the morning when participants are likely to be calmest and may have been less towards the end of the day.

Such interventions are the future whether we like it or not, and they may be more effective than traditional teaching and teaching methods and have a significant impact on many learners worldwide, in a variety of age-groups, settings and subject areas, ultimately enhancing the quality of learning and life experience for many.

## Figures and Tables

**Figure 1 behavsci-13-00023-f001:**
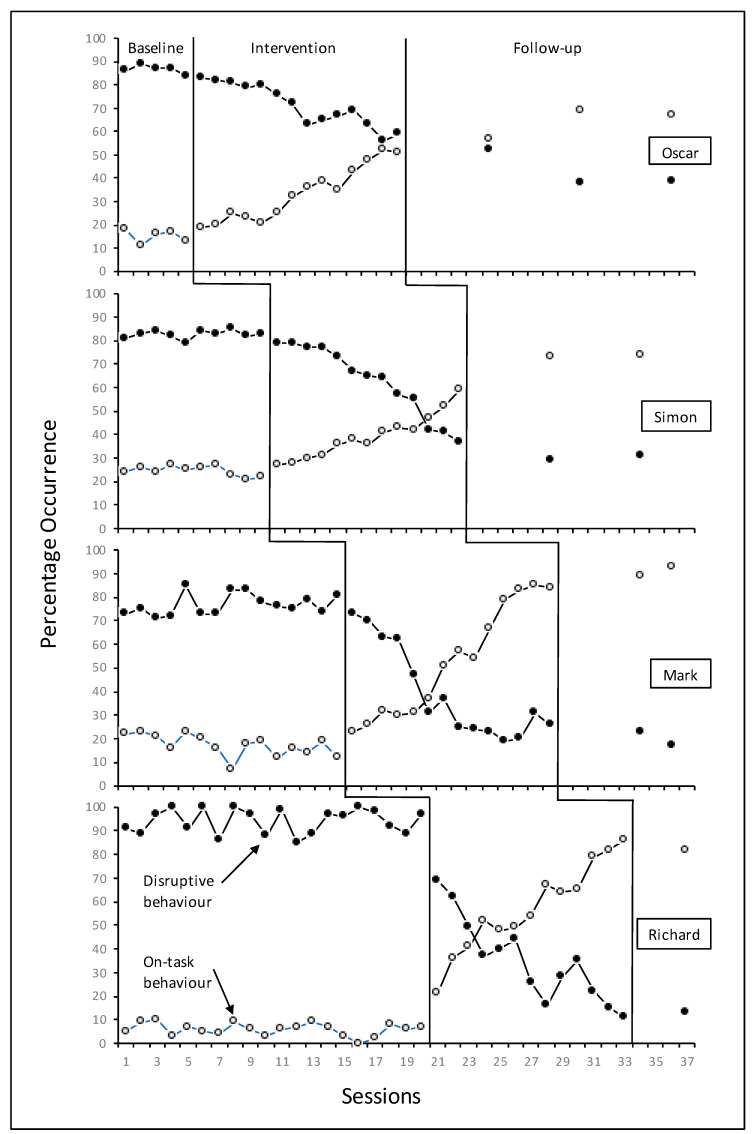
Percentage of 10 s intervals of on-task and disruptive classroom behaviours during 20 min classroom observations across all phases.

**Figure 2 behavsci-13-00023-f002:**
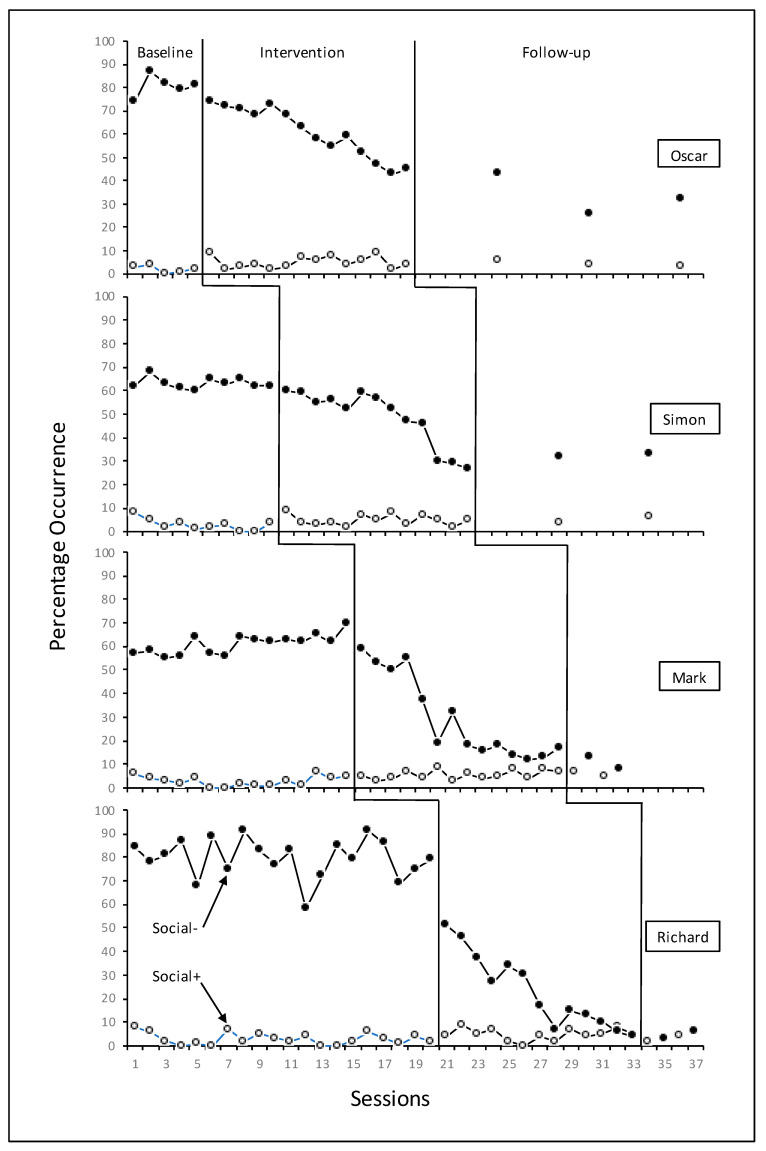
Percentage of 10 s intervals of teacher social contact during 20 min classroom observations across all phases.

**Table 1 behavsci-13-00023-t001:** On-task and disruptive classroom behaviours.

On-Task Behaviours	
1	Listen to teacher’s instructions (the participants oriented toward the teacher with eyes open and not talking).
2	Raising hand (left or right).
3	Writing on their notebook or booklet.
4	Reading written work (own quiet reading or loud to the class).
5	Talking to the teacher.
6	Discussing set work with a neighbour.
7	Having the teacher check work.
8	Talking to a group in a group discussion task.
9	Reading from the whiteboard.
10	Researching on the computer.
11	Looking up words in the computer/hard copy dictionary.
12	Complying with other verbal instructions given by the teacher/assistants.
13	Look up the 60” flat screen.
14	Disruptive behaviours
15	Noncompliance.
16	Yelling.
17	Out of seat behaviours included students sitting on their feet, standing up, lying down, and moving locations and seat without permission.
18	Making inappropriate noises included any vocal noise when the teacher had not indicated that the student may speak.
19	Getting out of seat.
20	Swinging on the chair.
21	Talking none relatedly to other students.
22	Making inappropriate contact with other students and objects included tapping other students, playing with other students’ hair, pushing other students with hands or feet, and touching materials not related to the current activity (e.g., nearby chairs, jewellery, shoelaces).

**Table 2 behavsci-13-00023-t002:** Social validity questionnaire.

#	Item	1	2	3	4	5
1	I find this App to be an acceptable way of increasing on-task and decreasing disruptive classroom behaviours.					
2	I would be willing for this App to be used again, to increase other appropriate and reduce other inappropriate classroom behaviours.					
3	I believe it would be acceptable to use this App without the consent of the participants.					
4	I like the App used in this study.					
5	I believe this App is likely to be effective in other educational environments and with other teachers, to increase on-task and decrease disruptive classroom behaviours.					
6	I experienced discomfort using this App.					
7	I believe this App is likely to result in permanent improvement in my teaching skills.					
8	I believe this App does not require many resources, and thus is affordable.					
9	Overall, I had a positive reaction to this method.					
10	Overall, I was satisfied from this App procedure.					

**Table 3 behavsci-13-00023-t003:** Participants’, teachers’, teaching assistants’ and parents’ responses to the use of the App.

Item #	I find this App to be an acceptable way of increasing on-task and decreasing disruptive classroom behaviours	I would be willing for this App to be used again to increase other appropriate and decrease other inappropriate classroom behaviours	I believe it to be acceptable to use this App without the consent of the participants	I like the App used in this study	I believe this App is likely to be effective in other educational environments and with other teachers, to increase on-task and decrease disruptive classroom behaviours	I experienced discomfort using this App	I believe this App is likely to result in permanent aaimprovement in my teaching skills	I believe this App does not require many resources, and thus is affordable	Overall, I had a positive reaction to this App	Everyone overall satisfaction from the entire App procedure
O	100	100	100	100	80	100	80	100	100	100
S	100	80	100	100	100	80	100	100	100	100
M	100	100	100	100	100	100	100	80	100	100
R	80	100	100	100	80	100	100	100	100	100
O’s Teacher	100	100	100	100	100	100	100	100	100	100
S’s Teacher	100	100	80	100	80	100	80	80	100	100
M’s Teacher	100	100	80	100	100	100	100	100	100	100
R’s Teacher	100	100	80	100	80	100	100	100	100	100
O’s Tea. Assis.	100	100	80	100	100	100	100	100	100	100
S’s Tea. Assis.	100	100	80	100	40	100	80	40	100	100
M’s Tea. Assis.	100	100	80	100	100	100	100	100	100	100
R’s Tea. Assis.	100	100	80	100	60	100	100	100	100	100
O’s Parents	100	100	60	100	100	100	80	100	100	100
S’s Parents	100	100	100	100	60	100	100	40	100	100
M’s Parents	100	100	80	100	100	100	100	100	100	100
R’s Parents	100	100	60	100	100	100	100	10	100	100

## Data Availability

Data is available from the first author.

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
