# Peer review of "High-Tech Classroom Management: Effects of the Use of an App on Disruptive and On-Task Classroom Behaviours for Students with Emotional and Behavioural Disorder"

_behavsci, 2022, doi:10.3390/bs13010023_

Round 1

Reviewer 1 Report

Dear authors,

thank you for a very interesting and enlightening reading!

From my side only three proposals for change and some remarks/questions: 

I suggest to delete (or update) the paragraph on development of smartphone use and app download - specifically the sentences string with: "Today, the number of smartphone users... and ending with "...(Arthur, 2014)" [in my copy line 123 to 133]. Data is developing so fast and numbers from 6 to 8 years ago, do not show a recent picture anymore. Even updated it would be outdated again in two years...

The description of the teacher training (2.4) could need some fine-tuning:  "approx. 120 minutes in seven steps" - indicates that it was one training session of approx. two hours, but steps 5-7 are on the next day. that could be clearer / And: explain who was doing the introduction/explanation for the participants (teacher?, researcher?, assistant?)

explain when and how (all groups together, separate...) "on-task and disruptive classrooms behaviour " above table 1)  was defined. It leaves a lot of questions on how the participants were involved, if they were empowered etc.

some additional remarks/questions:

also in the paragraph smartphone and app use: the last sentence refers to "researchers" (... increasing number of researchers appear to...). Is this correct? Or should it be teacher / lecturer / instructor?

And one thing that is very important for me, but maybe it is due only to my lack of competence regarding comma, and everything is ok:

(line 284 ff) the calculation of percentage of occurences (X) of on-task or disruptive behaviour - I read your sentence - because of the commas - as:

X = ((intervals on-task/possible intervals)-missed observations)*100

but I think it should be:

X =( (intervals on-task)/(possible intervals - missed observations) )*100

So I think, the comma behind intervals in: "dividing by the total number of possible intervals, minus any missed observations ...." should be deleted and a formula would definitely make it easier to read.

last remark an soft critique: using the same questionnaire on social validity for teachers, teaching assistants, and parents is ok, but for the 11 and 12 year old participants (with EBD and reported low intelligence) this might be overwhelming.

all the best!

Author Response

Dear reviewer 1. 

Thank you for your precious comments. 

Comment 1 - revised/updated and a new reference is provided.

Comment 2 - updated and revised as instructed.

Comment 3 - 'some additional remarks/questions..." - changed to teachers - thank you :)

Comment 4 - "And one thing that is very important for me..." - comma was taken out - true - thank you :).

Thank you for all your comments.

Reviewer 2 Report

The research is very interesting and relevant.

The introduction and justification are very well argued and with references to studies and research of great impact. However, the format of the appointments is not correct.

Citations and references are not in the mdpi format required by the journal. They must be exchanged. The final references are arranged in alphabetical order and following an APA standard that is not the one requested in this journal.

The justification of how an application that works what is proposed here should be, should appear at the end of section 1. But its description and characteristics, which also appear at the end of section 1, should be indicated in section 2. Method.

This research consists of a specific case study of four students, and should be indicated as such.

Likewise, in the conclusions, mention should be made of the limitations that this study presents (which are not few, and which among others are: a very small sample of 4 students that cannot give us significant results; the fact that the intervention with The students have done it first thing in the morning, which is the calmest they tend to be, and not that it would have been done towards the end of the school day, which is when there is a greater probability that they will show their fears and their states of anxiety; etc. .)

Author Response

Dearest second reviewer,

I thank you so much your your precious review of which I took careful steps to embed and respond to them in the manuscript - thank you.

  1. Citations were all revised to meet MDPI Journal requirement.
  2. Justification of the App in section 1 and 2 - revised and updated.
  3. "This research consists of a specific case study of four students" - updated and inserted in the study.
  4. "Likewise, in the conclusions, mention should be made of the limitations that this study presents (which are not few, and which among others are: a very small sample of 4 students that cannot give us significant results; the fact that the intervention with The students have done it first thing in the morning, which is the calmest they tend to be, and not that it would have been done towards the end of the school day, which is when there is a greater probability that they will show their fears and their states of anxiety; etc. .)" - revised and was added in the last section - Discussion - thank you.

    Once again, thank you very much for your important comments all of which I revised.

Best wishes.

Reviewer 3 Report

Strengths:

1. The paper demonstrates an exploratory investigation of the effects of the app developed on Disruptive and On-task Behavior for students with EBD. This is unique as most apps developed in education are mostly for students who do not require special needs. This also provides important clues on further experimental analysis that can be conducted in this field.

2. The paper is well-written. Objectives, methods, and results are clearly discussed. The problem is clearly identified based on the gaps found in the literature. Methods are clearly and succinctly discussed which allows other researchers to replicate and evaluate the scientific rigor.

3. Experimental design utilized in the study is highly appreciated as most studies involving EBDs utilize qualitative research design.

4. No corrections in the English language.

Weaknesses:

1. There is no control group for comparison on the effects thus the effects uncovered might not be due to the introduced intervention. Although this can be further discussed as part of the limitations.

2. On section 2.8, I suggest that the authors give a background of the social validity questionnaire and provide a discussion on which part(s) were revised. Was there reliability testing and validation conducted on this instrument?

3. There is no discussion on ethical considerations. This is important as the study involves children with special needs.

4. There is no disclosure of the application developed. This may be provided (if the intellectual property process will not be affected) as this will provide significant inputs on what features of the application may have contributed to the observed effects. Would it be possible to add a discussion on the features of the app developed? This would allow readers to have a picture of which aspect of the app may have been the reason for the positive result shown in the study.

5. In terms of engagement with scholarly sources, the authors may omit references that are older than 10 years and add more discussion on what has been done to address the problem at hand.

6. Why was the intervention not done at the same time? Please provide further discussion and justification on this experimental decision.

Author Response

Dearest Third Reviewer,

I would like to thank you for your precious comments. Your input is greatly appreciated.

Thank you for your first four comments.

Weaknesses:

  1. True - no control group yet - it is a single subject design utilizing 4 participants and 4 teachers all in multiple base-line design which enhance both internal and external experimental control and some limitations are discussed - thank you for noticing it :-).
  2. Revisions were made to section 2.8 and an explanation and discussion were added.
  3. Ethics - super important - I agree - it is discussed in section 2.1 towards the end - sorry and once again, thank you for highlighting this so important issue.
  4. An extended description of the App is provided in section 2.3. With that, and due to its innovative patent, further analysis may threat the intellectual property process as this App was developed specifically for the purpose of this study. Yet, a factor analysis assessing the various parts of the App ay have indicated, as you state in your wise comments, which specific aspect of the App was more responsible for the behavior change. This and other limitations were discussed in section 4. Thank you.
  5. Scholarly - revised and updated. Thank you.
  6. Explanation on why the intervention was not done at the same time - This study used multiple baseline design (see further explanation in section 2.5) to establish experiential control - both internal and external validity (further explanation in section 2.9). Because the size group is relatively small - unlike group design where we tend to examine large group of individuals (thousands etc) - in MBD we time the application of the intervention whilst maintaining baseline measures throughout. Then we assess within each participant the impact of the intervention on the behavior of each participant whilst continue to apply the intervention to the other participants. This method of delayed application of the intervention enables us to increase both internal validity and external as in essence we replicate the application on each participant. When assessing the data points on the graph (visually) we can tell whether or not experimental control is shown. This was discussed in section 4. 

I wish to again thank you for your so important comment and asking for further clarification.

Best wishes.